**Subject Category:**
Biology (whole organism)

palaeontology

Macropodoidea, Balbaridae, *Nambaroo*, *Dendrolagus*, gait evolution, Miocene

**Author for correspondence:**
Benjamin P. Kear
e-mail: benjamin.kear@em.uu.se

# Climbing adaptations, locomotory disparity and ecological convergence in ancient stem 'kangaroos'

Wendy Den Boer[1,2], Nicolás E. Campione[2,3,4]
and Benjamin P. Kear[5]

[1]Department of Palaeobiology, Swedish Natural History Museum, 104 05 Stockholm, Sweden
[2]Department of Earth Science, Uppsala University, Villavägen 16, 752 36 Uppsala, Sweden
[3]Department of Organismal Biology, Uppsala University, Norbyvägen 18A, 752 36 Uppsala, Sweden
[4]Palaeoscience Research Centre, School of Environmental and Rural Sciences, University of New England, Armidale, New South Wales 2531, Australia
[5]Museum of Evolution, Uppsala University, Norbyvägen 16, 752 36 Uppsala, Sweden

BPK, 0000-0002-3128-3141

Living kangaroos, wallabies and rat-kangaroos (Macropodoidea) constitute the most ecologically diverse radiation of Australasian marsupials. Indeed, even their hallmark bipedal hopping gait has been variously modified for bounding, walking and climbing. However, the origins of this locomotory adaptability are uncertain because skeletons of the most ancient macropodoids are exceptionally rare. Some of the stratigraphically oldest fossils have been attributed to Balbaridae—a clade of potentially quadrupedal stem macropodoids that became extinct during the late Miocene. Here we undertake the first assessment of balbarid locomotion using two-dimensional geometric morphometrics and a correlative multivariate analysis of linear measurements. We selected the astragalus and pedal digit IV ungual as proxies for primary gait because these elements are preserved in the only articulated balbarid skeleton, as well as some unusual early Miocene balbarid-like remains that resemble the bones of modern tree-kangaroos. Our results show that these fossils manifest character states indicative of contrasting locomotory capabilities. Furthermore, predictive modelling reveals similarities with extant macropodoids that employ either bipedal saltation and/or climbing. We interpret this as evidence for archetypal gait versatility, which probably integrated higher-speed hopping with slower-speed quadrupedal progression and varying degrees of scansoriality as independent specializations for life in forest and woodland settings.

# 1. Introduction

Crown macropodoids and their stem relatives delimit the broader clade Macropodiformes [1], which encompasses one of the most iconic radiations of marsupials. Today, they occupy a wide range of habitats from rainforests to deserts throughout Australia, New Guinea and adjacent islands [2–4]. This ecological variability is reflected in macropodoid locomotive disparity, which quintessentially incorporates bipedal hopping as the highest-speed gait in almost all extant species. The only exception is the morphologically primitive rainforest-dwelling Musky rat-kangaroo, *Hypsiprymnodon moschatus*, which persistently employs quadrupedal bounding involving synchronous use of the hind limbs [5,6]. Asynchronous walking and climbing have also developed secondarily [7] in the highly specialized tree-kangaroos referred to the genus *Dendrolagus* [8]. In addition, many larger-bodied kangaroos (as well as the atypical forest-dwelling *Dorcopsis* wallabies [7]) use pentapedal progression, integrating the muscular tail for stabilization and propulsion [9]. Pentapedal locomotion is specifically correlated with increased hind limb length and the preferential occupation of open habitats [10]. On the other hand, smaller-bodied wallabies and rat-kangaroos living in dense vegetation or on rocky outcrops tend to use bounding gaits at slower speeds [10,11].

The reconstructed locomotory habits of extinct macropodoids are similarly varied and have been mainly described among Pliocene–Pleistocene taxa. Examples include the large-bodied tree-kangaroo *Bohra* [12,13], and semi-arboreal macropodine *Congruus* [14], the gigantic short-faced sthenurines, which might have employed upright walking [15], as well as the potentially quadrupedal 'giant wallaby' *Protemnodon* [15,16], and cursorial 'giant rat-kangaroo' *Propleopus* [17]. By contrast, the postcranial remains of stratigraphically much more ancient stem macropodoids are substantially less completely documented. The oldest recorded specimens comprise a few isolated tarsal and pedal elements, together with a handful of incomplete skeletons from upper Oligocene [18,19] and Miocene strata [20–27]. The most morphologically peculiar of these have been referred to the basally branching clade Balbaridae [21,26–29] and are epitomized by a single articulated skeleton—the holotype (Queensland Museum, Brisbane, Australia [QM] F34532) of *Nambaroo gillespieae* [26]—which was recently assigned to another closely related genus *Ganawamaya* [30]. This fossil is distinguished by numerous osteological adaptations that indicate habitual quadrupedal locomotion, coupled with either bounding or hopping, and limited climbing ability [26]. The most notable of these features include a short tibia-fibula contact and reduced lateral trochlear crest on the astragalus (=talus [13]), which imply a capacity for rotational movement in the lower leg and upper ankle [26]. In addition, the expanded astragalar head/navicular facet and opposable first toe suggest that the pes was laterally flexible and capable of grasping [26]. Collectively, these character states are manifested elsewhere only in *H. moschatus*, which has likewise been observed to climb across fallen branches and tree trunks [5]. However, other balbarid postcranial remains referred to a separate taxon, *Balbaroo nalima*, display characteristics that are more consistent with higher-speed hopping [27], thereby implying locomotory variability within the clade. The most conspicuous traits are the prominent muscle scars for the *m. quadratus femoris* and adductor complex on the femoral shaft, and a deeply 'stepped' calcaneum-cuboid facet. This diagnostic tarsal configuration would have served to restrict lateral rotation of the foot [31,32] and is universal among macropodoids [6,29,32,33] (albeit secondarily reduced in *Dendrolagus* and *Bohra* [12,13]), including all ancient stem taxa for which sufficient postcranial bones are known [19,25–27].

The first quantitative assessment of balbarid pedal morphology was undertaken using a multivariate analysis of linear measurements derived from calcanea [34]. This revealed a broad distribution of balbarid morphotypes among modern bounding, bipedal saltating and climbing macropodoids, prompting the conclusion that all were likely capable of hopping, and some may have perhaps been arboreal [34]. Conflictingly, though, the obligate quadruped *H. moschatus* was also placed among hopping macropodoids (despite earlier analyses of calcaneal metrics suggesting closer proportional similarity to arboreal/scansorial marsupials [35]), possibly because it possesses traits that are indicative of terrestrial habits rather than hopping *per se* (e.g. an anteroposteriorly protracted calcaneal tuber calcis and transversely narrow calcaneum-astragalus articulation, which differ from the morphologies evident in tree-kangaroos [34]). Here we test the potential for hopping in balbarids, and infer their locomotory range, through a series of comparative multivariate analyses utilizing two-dimensional geometric morphometrics, together with a series of standard linear measurements. Our analyses build on previous studies [15,34,35], but alternatively focus on the astragalus and pedal digit IV ungual as independent proxies for primary gait. We selected these elements because they are preserved in the articulated holotype skeleton of *N. gillespieae* [26], as well as some other unusual

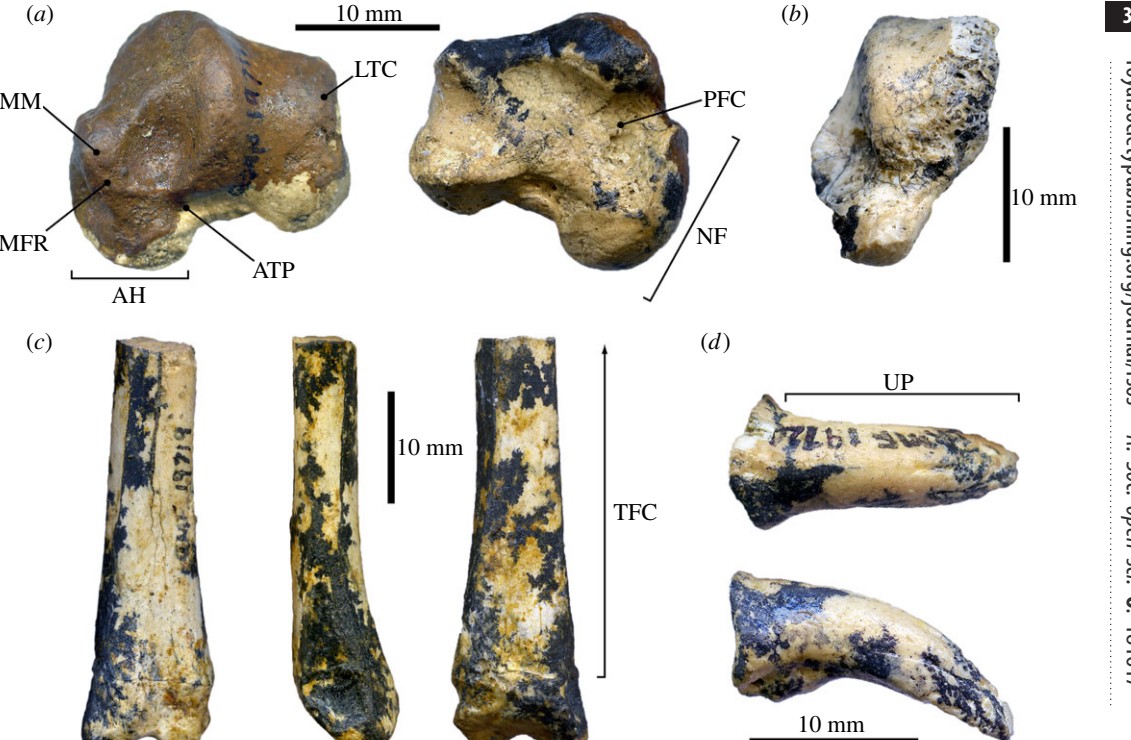

**Figure 1.** Early Miocene balbarid-like macropodoid postcranial elements recovered from 'Upper Site' in the Riversleigh World Heritage Area of Queensland, Australia. (*a*) QM F59022. (*b*) QM F59023. (*c*) QM F59024. (*d*) QM F59025. AH, astragalar head; ATP, astragalotibial 'pit'; LTC, lateral trochlear crest; MFR, malleolar fossa rim; MM, medial malleolus; NF, navicular facet; PFC, plantar facets for the calcaneum, TFC, tibia-fibula contact; UP, ungual process. Imaging by B.P.K.

balbarid-like hind limb remains that were recovered from an early Miocene deposit in the famous Riversleigh World Heritage Area (RWHA) of northwestern Queensland, Australia [36]. These new fossils exhibit derived character states that occur only in modern tree-kangaroos, and therefore provide novel evidence of independently specialized morphologies consistent with climbing habits in ancient stem macropodoids.

# 2. Material and methods

## 2.1. Study material

The macropodoid fossils described herein (figure 1 and electronic supplementary material, description of fossils and table S1) were identified in 1998 among jumbled bones recovered as a residue after the dissolution of a block of freshwater limestone in weak acetic acid (work undertaken by staff at the University of New South Wales, Sydney, Australia). This bulk matrix sample derived from the early Miocene 'Upper Site' (Faunal Zone interval B3 [37]) in the RWHA (see assemblage list in Archer *et al.* [38]). Anatomical associations between the different elements were lost either prior to burial, or during preparation, with further fragmentation and damage incurred via field extraction and transport to the laboratory. Despite this, morphological, ontogenetic and preservational compatibility suggests that these remains derive from at least two individuals of one or more closely related taxa. The specimens have been accessioned into an appropriate state museum repository with the following registration numbers (see electronic supplementary material for details on fossil cataloguing): QM F59022, a complete left astragalus; QM F59023, fragment of a left astragalus comprising the astragalar head, malleolar fossa and medial trochlear crest; QM F59024, the distal section of a left fibula incorporating the fused epiphysis; QM F59025, a pedal digit IV ungual phalanx with its proximal articular surface sheared off.

## 2.2. Geometric morphometrics

We analysed our two-dimensional digital landmark data using *geomorph* 3.07 [39] and standard statistical packages in *R* v.3.5.2 [40]. Original photographs of 30 astragali and 34 pedal digit IV unguals were

selected to represent the extant arboreal species of *Dendrolagus*, the obligate quadruped *Hypsiprymnodon moschatus* and various bipedal saltating macropodoids (electronic supplementary material, tables S2 and S3), together with the extinct balbarid *Nambaroo gillespieae* (QM F34532 found at the early Miocene 'Quantum Leap Site' and assigned to faunal zone intervals B2/B3 [37] of the RWHA), and the specimens QM F59022 and QM F59025. These were all digitized in *tpsDIG* v2.16 [41] with astragalar orientations standardized to the left-hand side, but with mirror imaging of bones from the right side when left-hand elements were not available; all left pedal digit IV unguals were photographed from the right-hand (medial) side. Inclusion of a 100 mm scale accommodated for slightly different imaging conditions. The outline of each astragalus was traced as an open curve in trochlear, medial, anterior, plantar, lateral and posterior views using the pencil tool in *tpsDIG*. These points were then resampled in *tpsDIG* with graphical editing in *tpsUTIL* v1.74 [42] and converted into 25 equidistant sliding semi-landmarks (electronic supplementary material, figure S1). The medial and lateral trochlear crests were also delimited by two fixed homologous landmarks placed at their extremities, and an additional semi-landmark marking the apex along an open curve; 10 equally spaced semi-landmarks defined the margins of the navicular facet. These same procedures were used to generate 20 equidistant semi-landmarks from open curves outlining the pedal digit IV ungual processes in dorsal and lateral views, with two homologous landmarks demarcating the ungual process base relative to the raised edges of the proximal articular surface and plantar process (electronic supplementary material, figure S1).

All morphospace distributions were visualized as bivariate plots derived from Principal Component (PC) analyses in *geomorph* with default settings and eigenvalues/eigenvectors calculated from the covariance matrix. Finally, a series of Binomial Logistic Regressions (BLRs) were run on a subset of the PCs (*glm* function) to assess the probability that our fossils could be assigned to either 'arboreal/scansorial', or 'terrestrial/bipedal saltating' ecology/locomotion categorization bins; these were defined using extant taxa of known habits [34,35] (electronic supplementary material, tables S2 and S3). *Hypsiprymnodon moschatus* was not assigned to any category because it is the only modern species that uses obligate quadrupedal gaits [5,6]. The probability of each category assignment was calculated by iteratively increasing the number of PCs until the data failed to produce a reliable BLR result. Preferred models were then chosen from among the convergent iterations based on their lowest Akaike Information Criterion (AIC) values. Almost all of our BLRs produced interpretable results, except for those using the astragalar plantar view dataset, which did not reach convergence. Our *R* code and tps/sliders data are included as electronic supplementary material for this article at the Dryad Digital Repository (https://datadryad.org/).

## 2.3. Linear morphometrics

Eight standardized linear measurements (electronic supplementary material, figure S1 and table S2) were taken from each astragalus in our sample using digital callipers. The parameters were specifically chosen to capture maximum anteroposterior length (ML) and width (MW), which reflect a relative projection of the astragalar head and medial malleolar process—both linked to intra-tarsal mobility [15]. Maximum depth on the lateral side from the apex of the trochlear crest (MD), and maximum height (HNF) and width of the navicular facet (WNF) constitute additional measures of changing rotational flexibility within the ankle, and between the pedal digits [12,13,15,16,32]. Lastly, the maximum length of the lateral (LLTC), and medial trochlear crests (LMTC), as well as maximum width of the articular sulcus (WTS) depict elongation of the trochlear articular surface, which reaches an acme in saltating macropodoids [12,13,15,16,31,32,43]. All linear measurements were $log_{10}$-transformed and subjected to a PC analysis in *R* [40]. BLRs were again run following the procedures outlined above, with the exception that PC1 was not considered because it is associated with changing body size.

# 3. Results

## 3.1. Anatomical comparisons

The fossils QM F59022–QM F59025 are morphologically comparable to other Oligocene–Miocene macropodoids with documented postcranial skeletons, and especially the balbarid *Nambaroo gillespieae*. This taxon possesses an identical (figure 1*a,b*) 'condyle-like' navicular facet, and a potentially synapomorphic inflated lateral rim enclosing the malleolar fossa on the astragalus [26]. Weak mediolateral compression of the distal shaft of the fibula (figure 1*c*) has likewise been reported in

*Balbaroo nalima* [27] (together with various stem macropodids and potoroids [19,24,25]), but contrasts with more crownward macropodids, such as *Rhizosthenurus flanneryi*, which has been interpreted as a bipedal saltator [23,34], or transitional asynchronous 'walker' [15]. Uniquely, however, QM F59022– QM F59025 exhibit multiple states that are usually indicative of tree-kangaroos (most notably the advanced fully arboreal species of *Dendrolagus* [13]): a proximodistally shortened tibia-fibula contact (figure 1*c*) [12,26]; mediolaterally broad astragalus accentuated by expansion of the astragalar head, neck and medial malleolus (figure 1*a,b*); sloping astragalar trochlear articular sulcus with an extremely reduced lateral trochlear crest (figure 1*a*); a very shallowly excavated astragalotibial 'pit' (= bursal/ trochlear notch [22,23,25,26,43,44]) on the dorsal surface of the astragalar neck (figure 1*a,b*); elongation and confluence of the plantar facets for the calcaneum (figure 1*a*); and in particular, the conspicuously oblique orientation of the navicular facet (observable in both QM F59022 and QM F59023: figure 1*a,b*), which differs from *N. gillespieae* and *Hypsiprymnodon moschatus* where the navicular articulations are vertically aligned relative to the dorsoventral axis of the medial trochlear crest [26]. The distinctively narrow, down-curved pedal ungual of QM F59025 (figure 1*d*) provides another distinguishing characteristic that resembles *Dendrolagus* and *Bohra* [45,46] but contrasts with *N. gillespieae* and *H. moschatus*, in which lateral compression of the ungual process is less pronounced, and thus more comparable to the extant rock-wallaby *Petrogale*, as well as other ecologically analogous taxa including the late Pliocene–early Pleistocene *Macropus mundjabus* [47].

## 3.2. Morphometrics

The results of our geometric morphometric analyses returned between 51 and 68% of the total variance explained by PC1 and PC2 in the astragalar landmark datasets (electronic supplementary material, table S4). In the trochlear, medial and anterior view analyses, these axes described the shape and proportional differences affecting the astragalar head, neck and medial malleolus, as well as the relative height of the medial trochlear crest, and both orientation and anteroposterior extent of the navicular facet (figure 2*a–c*). Peripheral outline of the plantar articular surface, particularly the transverse width of its posterior edge, and medial expansion of the navicular facet articular surface, together with the relative heights of the lateral versus medial trochlear crests, were additionally depicted by the plantar, lateral and posterior view analyses (figure 3*a–c*).

The pronounced medial malleolar process and 'condyle-like' navicular facet of QM F59022 and *Nambaroo gillespieae* produced negative loadings for these taxa along PC1 in the trochlear, medial and anterior view analyses (figure 2*a–c*). Likewise, the posterior extent of the lateral trochlear crest and shallowing of the trochlear articular sulcus (concomitant with the reduction of the medial trochlear crest) generated extreme negative loading of QM F59022 and *N. gillespieae* along PC1 in the posterior view analysis (figure 3*c*). By contrast, variation in the relative transverse width of the posterior edge of the astragalus yielded negative placement of QM F59022 on PC1 in plantar view, but alternatively generated positive loading of *N. gillespieae* along this same axis (figure 3*a*). Anterior extension of the lateral trochlear crest positively loaded both QM F59022 and *N. gillespieae* along PC1 in the lateral view analysis (figure 3*b*), but otherwise segregated these specimens on PC2, where the shorter, more arched profile of the lateral trochlear crest in QM F59022 is reflected by its positive loading. Lastly, both QM F59022 and *N. gillespieae* were positively positioned on PC2 in the trochlear, anterior and plantar view analyses, which captured elongation of the astragalar neck (figure 2*a*), in addition to increasing height of the medial trochlear crest (figure 2*c*), and medial deflection of the navicular facet (figure 3*a*); these morphologies were also represented by negative loading of these specimens on PC2 in the medial (figure 2*b*) and posterior view analyses (figure 3*c*).

Although QM F59022 and *N. gillespieae* frequently plotted outside the morphological range of most extant macropodoids (see figures 2*a,b* and 3*a,c*), a few modern species were found to occupy immediately proximal areas of morphospace. In particular, the obligate quadruped *Hypsiprymnodon moschatus* (figures 2*a* and 3*c*) exhibited a comparably narrow trochlear articular surface, together with posteriorly extensive medial trochlear crest, and expansive medial malleolar fossa [26]. Anterior projection of the astragalar neck and increasing width of the navicular contact were also evident in some species of the tree-kangaroo *Dendrolagus* (figure 2*b*), as well as potoroines (e.g. *Potorous*, *Bettongia*: figures 2*b,c*), which are known to frequently employ slow quadrupedal progression [7,11]. In addition, various smaller-bodied macropodids displayed very similar astragalar morphologies (figures 2*a,c* and 3*b*), especially those that tend to use quadrupedal bounding and/or hopping to manoeuvre through spatially complex habitats, such as dense vegetation and rocky uneven terrain (e.g. *Setonix*, *Petrogale* [7,10,11,48]).

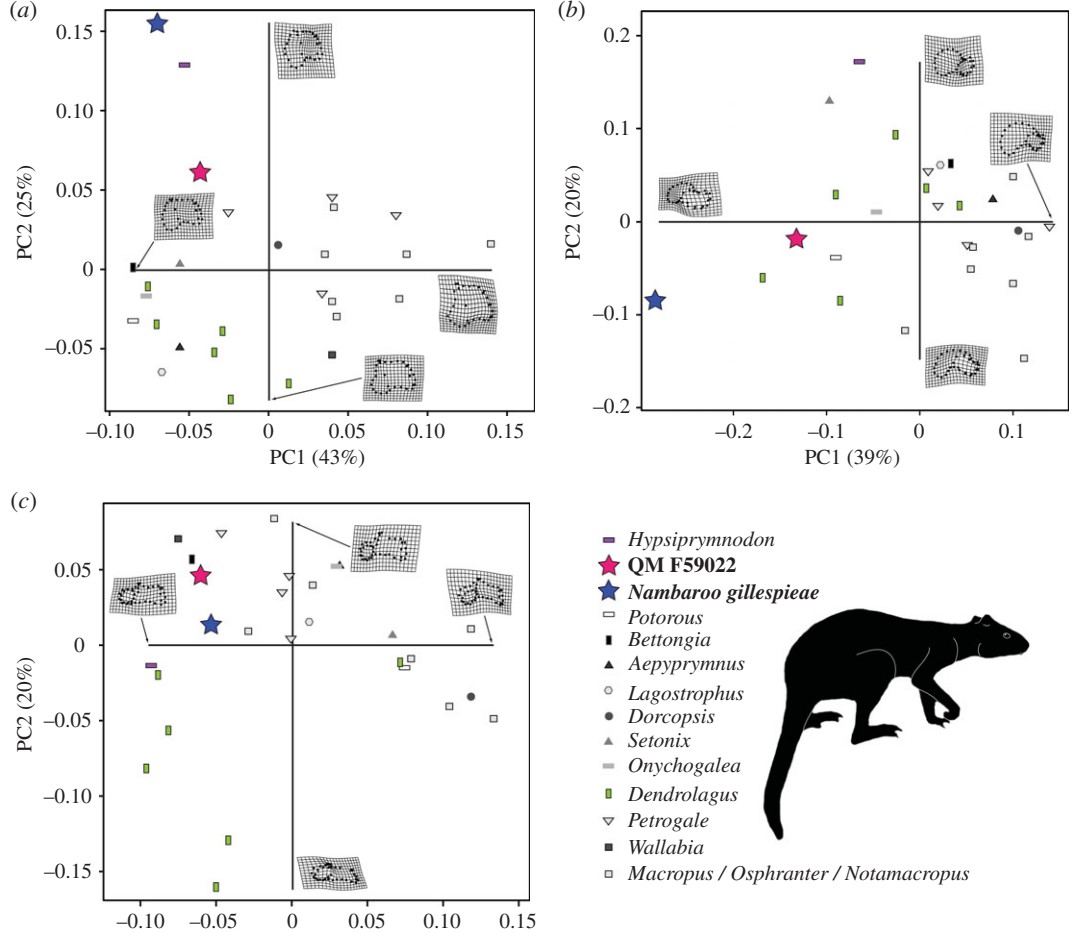

**Figure 2.** Geometric morphometrics of the astragalar landmarks datasets. Results depict PC1 and PC2 derived from the (*a*) trochlear, (*b*) medial and (*c*) anterior view analyses. Morphospace distribution shows QM F59022 (red star) and *Nambaroo gillespieae* (blue star; bottom right body-form silhouette) relative to the extant quadruped *Hypsiprymnodon moschatus* (purple horizontal rectangle), the arboreal species of *Dendrolagus* (green vertical rectangles) and bipedal saltating crown potoroines and macropodines (various greyscale shapes). Interpretations of morphology along the axes are depicted as thin-plate spline (*tps*) deformation grids. Graphics produced with *R* and *Adobe* CC2016 by N.E.C and B.P.K.

The results of our astragalar linear measurements dataset yielded 85% of the total variance within the size-related vector PC1 (electronic supplementary material, table S5). On the other hand, PC2 and PC3 derived 9% of their correlated variable loadings from proportional differences in WTS and ML versus HD and HNF, and WNF versus MW, LMTC and LLTC, respectively (figure 4*a*). QM F59022 and *N. gillespieae* were found to segregate along both PC2 and PC3, particularly in relation to WTS, which is conspicuously reduced in QM F59022 (see description in the electronic supplementary material), and WNF, which is compatible with *Dendrolagus* [45,46]. By contrast, both WNF and HNF separated *N. gillespieae* as an extreme outlier, together with some other proportionately comparable taxa, including *H. moschatus* [26].

Morphological segregation of QM F59025 and *N. gillespieae* was further evidenced in our plots of the pedal digit IV ungual landmark datasets (figure 4*b,c*). These yielded either 86% or 75% of the PC1/PC2 cumulative variance from the dorsal and lateral views, respectively (electronic supplementary material, table S6). PC1 depicted increased tapering and downward curvature of the ungual process, while PC2 captured relative constriction of the ungual process along its midsection. The morphospace placement of QM F59025 approached that of *Dendrolagus*, thus confirming our anatomical observations. Conversely, the broader, straighter profile of the pedal claws in *N. gillespieae* more closely resembled *H. moschatus*, and some other larger-bodied saltating macropodines, such as *Petrogale* and species of *Notamacropus*.

Our BLR ecology/locomotion predictions from the astragalar trochlear, anterior and lateral view datasets (figure 5*a,c,d*), as well as the astragalar linear measurements (figure 5*f*), and pedal digit IV

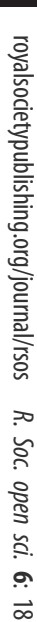

**Figure 3.** Geometric morphometrics of the astragalar landmarks datasets. Results depict PC1 and PC2 derived from the (*a*) plantar, (*b*) lateral and (*c*) posterior view analyses. Morphospace distribution shows QM F59022 (red star) and *Nambaroo gillespieae* (blue star) relative to the extant quadruped *Hypsiprymnodon moschatus* (purple horizontal rectangle), the arboreal species of *Dendrolagus* (green vertical rectangles; bottom right body-form silhouette) and bipedal saltating crown potoroines and macropodines (various greyscale shapes). Interpretations of morphology along the axes are depicted as *tps* grids. Graphics produced with *R* and *Adobe* CC2016 by N.E.C. and B.P.K.

ungual lateral view dataset (figure 6*b*), all advocated morphological categorization of QM F59022, QM F59025 and *N. gillespieae* with terrestrial/bipedal saltating macropodoids. This result is significant because the residual variance from these analyses was found to diminish substantially relative to the original null deviance, denoting their robust predictive potential (table 1). Nevertheless, the astragalar medial and posterior views, together with digit IV ungual dorsal view dataset indicated arboreal/ scansorial habits, especially for QM F59022 and QM F59025 (figures 5*b*,*e* and 6*a*), which not only demonstrates ambiguity in our ability to precisely estimate ecological and locomotory capabilities but also highlights the considerable novel anatomical variation exhibited by these ancient stem macropodoids, even within single bones.

# 4. Discussion and conclusion

Although QM F59022–QM F59025 share multiple synapomorphies with Balbaridae, we have deferred a formal classification herein because these remains cannot yet be distinguished from those of the closely related Propleopinae (extinct 'giant rat-kangaroos') [1,19,26–30], whose postcranial osteology is virtually unknown [17]. Representatives of both these clades occur coevally at 'Upper Site' [37], as well as other late Oligocene–Miocene deposits throughout the RWHA [49] and have similar estimated body masses [49]. In addition, QM F59022–QM F59025 exhibit character states that are directly comparable with the scansorial macropodines *Dendrolagus* and *Bohra* [13]. However, the proposed late Miocene divergence timeframe for *Dendrolagus* [50], in conjunction with the early–middle Pliocene to

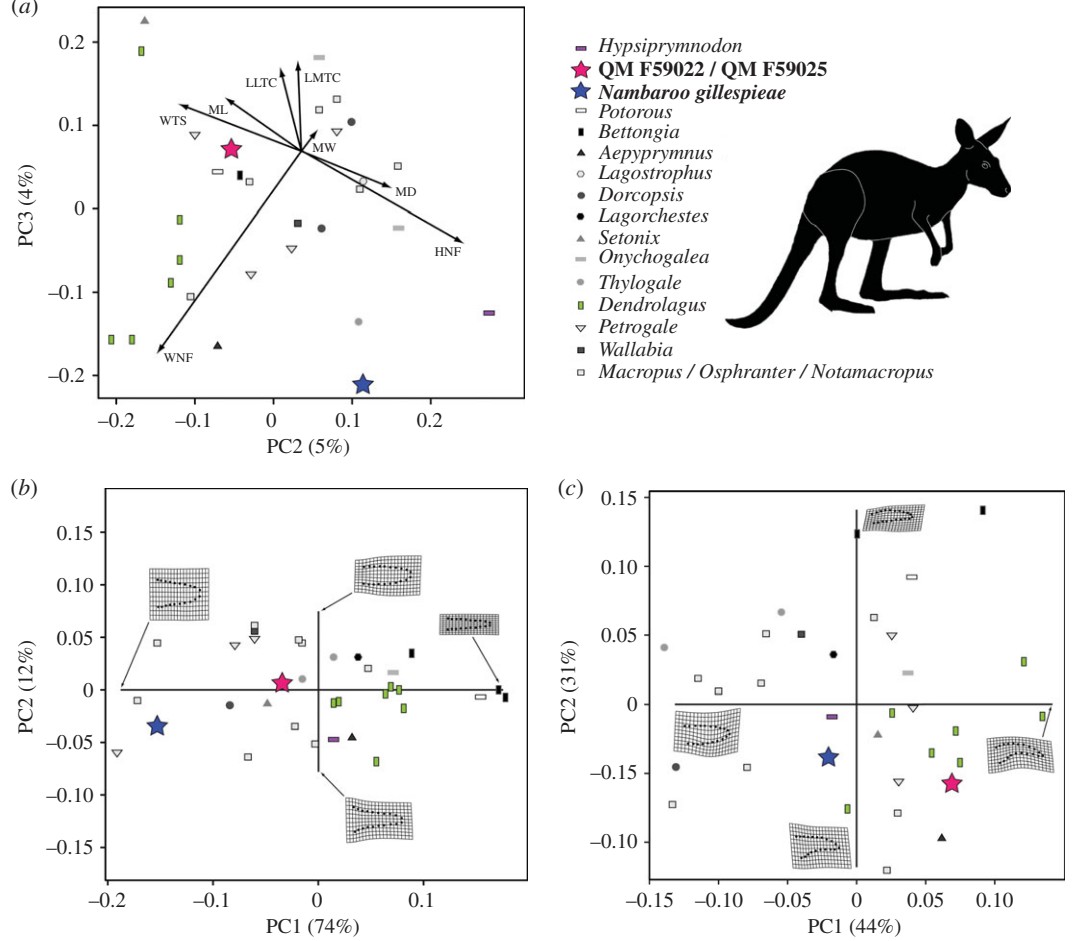

**Figure 4.** Linear morphometrics of the astragalar linear measurements (*a*), and geometric morphometrics of the pedal digit IV ungual (*b*) dorsal and (*c*) lateral view landmark datasets. Linear measurement results depict PC2 and PC3 (PC1 was excluded as the size dependent variable). Landmark results visualize PC1 and PC2. Morphospace distributions show QM F59022 (red star) in (*a*), and QM F59025 (red star) in (*b,c*), together with *Nambaroo gillespieae* (blue star) relative to the extant quadruped *Hypsiprymnodon moschatus* (purple horizontal rectangle), the arboreal species of *Dendrolagus* (green vertical rectangles), and bipedal saltating crown potoroines and macropodines (various greyscale shapes; top right body-form silhouette). Interpretations of morphology along the landmark axes are depicted as *tps* grids. Graphics produced with *R* and *Adobe* CC2016 by N.E.C and B.P.K.

Pleistocene [12,45,46,51] stratigraphic distributions of *Dendrolagus* and *Bohra* fossils (together with the semi-arboreal *Congruus* [14]), argues against our specimens evidencing a macropodine tree-kangaroo ghost lineage. Rather, we conclude that QM F59022–QM F59025, together with more confidently assigned balbarid specimens [26,34], collectively evince convergent adaptations in their hind limbs that enabled climbing. This hypothesis is supported by current macropodiform phylogenies (figure 7), which place *Hypsiprymnodon moschatus* as the most basally branching crown macropodoid [50], and probable stemward relative of both propleopines and balbarids [1,30]. *Hypsiprymnodon moschatus* is a relic species that retains an antecedent terrestrial lifestyle but is also able to climb [5,6]. The far more derived scansorial characteristics evident in the limb and pedal bones of QM F59022–QM F59025 thus demonstrate an independent elaboration of these capabilities in early Miocene macropodoids, which predate the oldest ancestors of *Dendrolagus* and *Bohra* by as much as 10 Ma (extrapolated from maximum stratigraphic ages [37] and divergence estimates [50]).

Previous assessments of functional anatomy [26,27] and calcaneal metrics [34] have posited contrasting capacities for bipedal saltation in different balbarid taxa. This accords with our morphospace plots and BLRs, which likewise suggest substantial variation. Nevertheless, such similitudes preclude any definitive inferences of primary locomotion, especially relative to extant macropodoids, which employ hopping, quadrupedal bounding and/or climbing to achieve a range of speeds and accommodate different habitat settings [7,11,48]. We, therefore, prefer to interpret our results as indicative of comparable gait versatility in ancient fossil macropodoids, which potentially

**Figure 5.** Box plots summarizing the Binomial Logistic Regressions (BLR) from our astragalar landmark (*a–e*), and linear measurement (*f*) datasets. Extant taxon ecology/locomotion categorization bins (*x*-axis) are plotted against ecology/locomotion gradients as predicted by the BLRs (*y*-axis). Ecology/locomotion BLR projections for QM F59022 (red dotted line) and *Nambaroo gillespieae* (blue dashed line) indicate morphological compatibility with both extant terrestrial/bipedal saltating (crown macropodines and potoroines delimited by grey fill; outliers = ◯) macropodoid species (*a,c,d,f*), and the arboreal/scansorial (delimited by green fill) tree-kangaroo *Dendrolagus* (*b,e*). Graphics produced with *R* and *Adobe* CC2016 by N.E.C. and B.P.K.

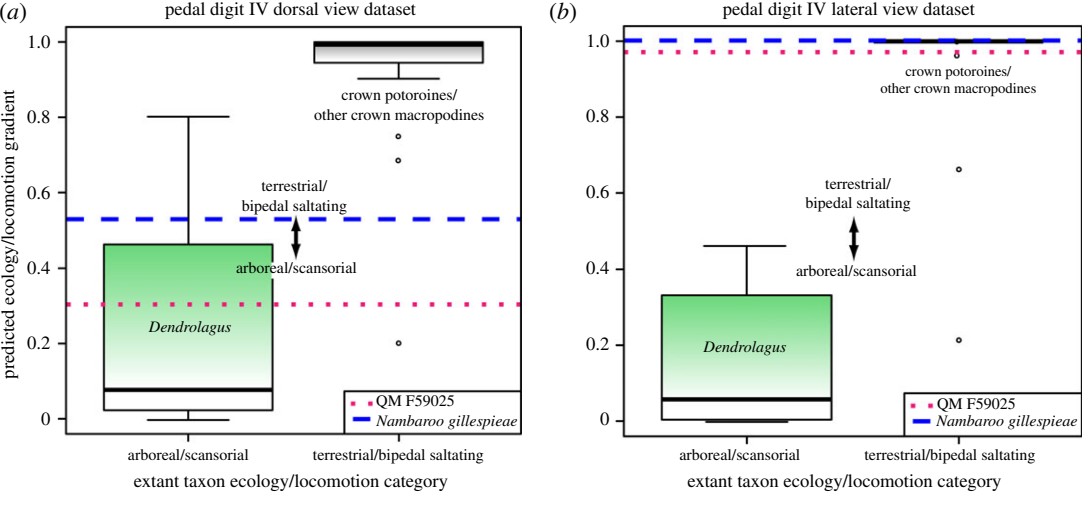

**Figure 6.** Box plots summarizing the BLR results from our pedal digit IV dorsal (*a*), and lateral (*b*) view landmark datasets. Extant taxon ecology/locomotion categorization bins (*x*-axis) are plotted against ecology/locomotion gradients as predicted by the BLRs (*y*-axis). Ecology/locomotion BLR projections for QM F59025 (red dotted line) and *Nambaroo gillespieae* (blue dashed line) indicate morphological compatibility with both extant terrestrial/bipedal saltating (crown macropodines and potoroines delimited by grey fill; outliers = ○) macropodoid species (*a*,*b*), and the arboreal/scansorial (delimited by green fill) tree-kangaroo *Dendrolagus* (*b*). Graphics produced with *R* and *Adobe* CC2016 by N.E.C. and B.P.K.

**Table 1.** Parameters, results and ecology/locomotion predictions generated by the Binomial Logistic Regression (BLR) analyses. Prediction probabilities: approaching 0 = arboreal/scansorial; approaching 1 = terrestrial/bipedal saltating. AIC, Akaike Information Criterion.

| dataset | PCs | variance (%) | AIC | deviance null | deviance residual | prediction *Nambaroo gillespieae* | QM F59022/ QM F59025 |
|---|---|---|---|---|---|---|---|
| astragalus landmarks | | | | | | | |
| trochlear | 1–4 | 77 | 17.1 | 27.55 | 7.07 | 1 | 1 |
| medial | 1–3 | 70 | 24.3 | 26.99 | 20.32 | 0.024 | 0.244 |
| anterior | 1–3 | 68 | 14 | 27.55 | 7.96 | 0.759 | 0.965 |
| lateral | 1–6 | 88 | 25 | 27.55 | 10.95 | 0.912 | 0.992 |
| posterior | 1–6 | 92 | 21.9 | 27.55 | 13.87 | 0.104 | 0.025 |
| astragalus linear measurements | | | | | | | |
| | 2 | 4.5 | 14.3 | 28.6 | 10.32 | 1 | 0.963 |
| pedal digit IV ungual landmarks | | | | | | | |
| dorsal | 1–6 | 99 | 25.6 | 33.12 | 11.61 | 0.53 | 0.306 |
| lateral | 1–5 | 97 | 19 | 32.6 | 7.01 | 1 | 0.971 |

employed higher-speed bipedal hopping and/or quadrupedal bounding, in conjunction with pentapedal progression or asynchronous walking at slower speeds. As in *H. moschatus* [5,6], scansoriality probably constituted an ancillary component in this gait repertoire for taxa such as *Nambaroo gillespieae*, but might have involved more active movement through the canopy vegetation in morphologically derived forms represented by QM F59022–QM F59025. Notably, sympatric species of *Dendrolagus* [8] and *Bohra* [12,13,45,46,51] also display varying propensities for climbing, and indeed, vertical partitioning of terrestrial versus arboreal habitats (as well as diurnal, crepuscular and nocturnal biorhythms [5,53]) might have been integral for early Miocene macropodoid communities, which reached their peak abundance in the humid forest palaeoenvironments of the RWHA [49].

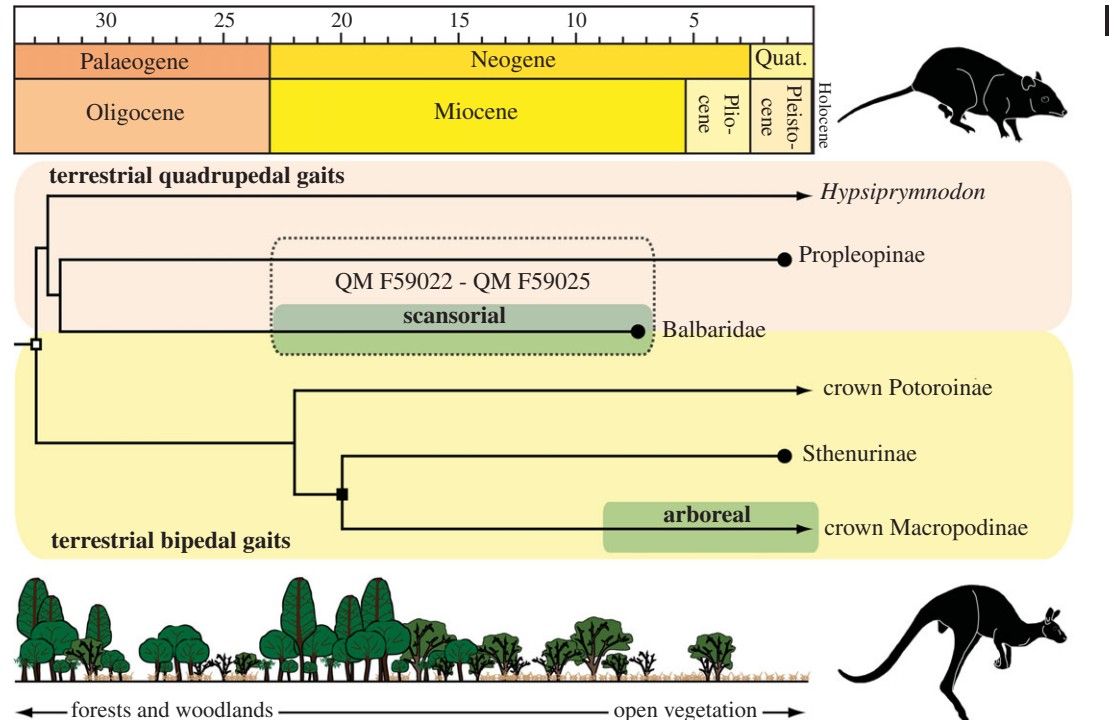

**Figure 7.** Phylogenetic timescale [1,51] of Macropodoidea (white square) depicting the convergent appearance of arboreal/scansorial locomotion (green) in balbarids and crown Macropodidae (black square). Background colour gradient infers the hypothesized locomotory transition [26] from predominantly terrestrial quadrupedal progression (tan), as characterized by the extant quadruped *Hypsiprymnodon moschatus* (top right body-form silhouette), to increasingly bipedal saltating gaits (yellow) in crown potoroines and macropodines (bottom right body-form silhouette); this is correlated against a schematic of changing palaeohabitats through time (modified from Kear *et al*. [52]). Dashed outline indicates uncertain taxonomic assignment of QM F59022−QM F59025. Graphics produced with *Adobe* CC2016 by B.P.K.

The final extinction of balbarids has been attributed to the onset of drier conditions during the late Miocene [54], or dietary competition with ancestral potoroids and macropodids that were better suited to processing fibrous xeric-adapted plants [49]. While our findings do not immediately elucidate either of these scenarios, they demonstrate that early Miocene balbarids and their relatives were ecologically diverse and had evolved derived scansorial morphologies that would have been ideal for life in forests and woodlands. Equivalent adaptations have not yet been identified in any crown-line Oligocene−Miocene macropodoid lineage [19,22−25], implying that while competitive exclusion might have been possible, it is unlikely to have impacted upon all stem macropodoid species to the same degree. Conversely, the progressive shift towards more open vegetation types that took place after the mid-Miocene Climatic Optimum [55,56] would have severely affected folivorous browsers [34], and perhaps restricted the available habitat of balbarids, which were already in significant decline [49]. Specialization towards scansorial lifestyles could therefore have proven similarly detrimental, but as revealed by our data (and that of others [34]), was nonetheless indicative of intrinsic locomotory versatility that has since contributed to the persistent survival of crown macropodoids and their antecedents in the face of changing environments.

Ethics. There were no Research Ethics required for this study. The fossils QM F59022−QM F59025 are permanently housed in the QM state repository.

Data accessibility. All datasets generated by this study are available from the Dryad Digital Repository: http://dx.doi.org/10.5061/dryad.m66b10q.2 [57].

Authors' contributions. W.D.B. and B.P.K. designed the study, collected/analysed the data, interpreted the results and wrote the manuscript. N.E.C. designed and ran the analyses, interpreted the results and wrote the manuscript. B.P.K. conceived and funded the project. All authors were involved in discussing the results and revising the manuscript.

Competing interests. We declare that we have no competing interests.

Funding. W.D.B. was funded by a Swedish Research Council grant (2011-3587) awarded to B.P.K. N.E.C. was supported by competitive funding to B.P.K. with co-financing from the departments of Earth Science and Organismal Biology at Uppsala University, Sweden.

Acknowledgements. Andrew Rozefelds and Kristen Spring (Queensland Museum), Sandy Ingleby and Anja Divljan (Australian Museum) and Karen Roberts (Melbourne Museum) generously provided access to specimens and photographs. Both the excavation and initial research loan of QM F59022–QM F59025 to B.P.K. at the South Australian Museum (Adelaide) were facilitated by Michael Archer and staff at the University of New South Wales (Sydney). Editorial comments from Andrew Dunn, Monica Daley and two anonymous reviewers improved our manuscript.

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
