## [Reviewer comments · Royal Society Open Science]

Review History

RSOS-181617.R0 (Original submission)

Review form: Reviewer 1

Is the manuscript scientifically sound in its present form?

Yes

Are the interpretations and conclusions justified by the results?

No

Is the language acceptable?

Yes

Is it clear how to access all supporting data?

Yes

Do you have any ethical concerns with this paper?

No

Have you any concerns about statistical analyses in this paper?

No

Recommendation?

Accept with minor revision (please list in comments)

Comments to the Author(s)

This manuscript aims to interpret the locomotor mode of early Macropodiformes from pedal morphology through morphometric comparisons with modern species. The authors examine some of the best preserved materials for stem macropodoids and use both traditional linear metrics as well as 2D geometric morphometrics of bones.

The analyses are clearly set out, and often do not give a clear answer to the questions asked. This may be because these earlier forms are less derived or more generalised than the majority of modern macropodoids. The key result is the BLR summary (Figure 5) that gives the mean position of both fossils at ~0.70-0.95 along the locomotor gradient towards the terrestrial/bipedal saltating end. However, the main conclusions in the text do not follow this finding, and suggest that arboreal/scansorial is more likely (L304, L331) and represents a convergent adaptation to arboreality (L37; Figure 6). The results and conclusions should more closely align to allow sufficient support. Statements such as 'archetypal gait versatility' (L35) seem to indicate that no firm conclusion can be drawn, and that they fit somewhere in between the extremes, and so the conclusion of convergent evolution is not warranted.

L32: 'distinct amongst' - 'distinct from' is clearer.

L77: change 'inferring' to 'implying'.

L136: does 'photographed from the right-hand side' refer to lateral or medial direction, as this depends on whether the bone is from the right or left side of the animal?

L157: the wording 'data became insufficient' is unclear given that you are increasing the number of PCs included in the analysis.

L183: references to figures would help here in describing the anatomical details.

L254: ML is nearly parallel to WTS in the figure, but is listed separately here. LLTC and LMTC fall intermediate to the other groups.

L257: does 'dimensionally compatible' mean approximately the same size?

L271: the latter part of this paragraph is more Discussion than Results.

L301: correct spelling of 'Hypsiprymnodon'. Also L303, L399.

L304: 'far more advanced scansorial ... QM' - the end of the Results, and data in Table 1, show that these fossils were as much 'terrestrial/bipedal saltating' than 'arboreal/scansorial', which calls into question this line of conclusions of arboreality, either primitively or independently evolved. Likewise L331 'advanced scansorial morphologies'.

L518: change to 'dependent'.

Review form: Reviewer 2**Is the manuscript scientifically sound in its present form?**

Yes

Are the interpretations and conclusions justified by the results?

Yes

Is the language acceptable?

Yes

Is it clear how to access all supporting data?

Yes

Do you have any ethical concerns with this paper?

No

Have you any concerns about statistical analyses in this paper?

No

Recommendation?

Accept with minor revision (please list in comments)

Comments to the Author(s)

Overall I believe this is a rigorous analysis and the manuscript is clear and well written. I have only a few relatively minor comments that I feel should be clarified.

1 - Many of the conclusions drawn from this study are supported by data in the supplementary files. Why were these figures/tables not included in the main manuscript?

2 - In figure 1, LTC is not defined.

3 - In figures 2-4, I was confused by the outlines of macropods adjacent to the figure legends. What is the significance of these?

4 - I expect that it will not be an issue in publication, but in my printed copy, the symbols in figures 2-4 were difficult to distinguish.

Decision letter (RSOS-181617.R0)

11-Dec-2018

Dear Dr Kear

On behalf of the Editors, I am pleased to inform you that your Manuscript RSOS-181617 entitled "Climbing adaptations, locomotory disparity, and ecological convergence in ancient stem 'kangaroos'" has been accepted for publication in Royal Society Open Science subject to minor revision in accordance with the referee suggestions. Please find the referees' comments at the end of this email.

The reviewers and handling editors have recommended publication, but also suggest some minor revisions to your manuscript. Therefore, I invite you to respond to the comments and revise your manuscript.

- Ethics statement

- Data accessibility

If you wish to submit your supporting data or code to Dryad (<http://datadryad.org/>), or modify your current submission to dryad, please use the following link:
<http://datadryad.org/submit?journalID=RSOS&manu=RSOS-181617>

- Competing interests

- Authors' contributions

- Acknowledgements

- Funding statement

Because the schedule for publication is very tight, it is a condition of publication that you submit the revised version of your manuscript before 20-Dec-2018. Please note that the revision deadline will expire at 00.00am on this date. If you do not think you will be able to meet this date please let me know immediately.

If your manuscript is newly submitted and subsequently accepted for publication, you will be asked to pay the article processing charge, unless you request a waiver and this is approved by Royal Society Publishing. You can find out more about the charges at

<http://rsos.royalsocietypublishing.org/page/charges>. Should you have any queries, please contact openscience@royalsociety.org.

on behalf of Dr Monica Daley (Associate Editor) and Kevin Padian (Subject Editor)
openscience@royalsociety.org

Associate Editor Comments to Author (Dr Monica Daley):

Associate Editor: 1

Comments to the Author:

Two expert reviewers have evaluated your paper interpreting locomotor modes of macropodoids from pedal morphometrics. The reviewers suggest that the work makes a scientifically sound contribution, but they also highlight the ambiguity of the results, and suggest that some of the conclusions drawn in the text should be revised to be better aligned with this ambiguity. I will consider a revised version of this paper that fully addresses the reviewers comments to make sure that all of the main findings and conclusions are fully supported by the data presented within the main text.

Reviewer comments to Author:

Reviewer: 1

Comments to the Author(s)

This manuscript aims to interpret the locomotor mode of early Macropodiformes from pedal morphology through morphometric comparisons with modern species. The authors examine some of the best preserved materials for stem macropodoids and use both traditional linear metrics as well as 2D geometric morphometrics of bones.

The analyses are clearly set out, and often do not give a clear answer to the questions asked. This may be because these earlier forms are less derived or more generalised than the majority of modern macropodoids. The key result is the BLR summary (Figure 5) that gives the mean position of both fossils at ~0.70-0.95 along the locomotor gradient towards the terrestrial/bipedal saltating end. However, the main conclusions in the text do not follow this finding, and suggest that arboreal/scansorial is more likely (L304, L331) and represents a convergent adaptation to arboreality (L37; Figure 6). The results and conclusions should more closely align to allow sufficient support. Statements such as 'archetypal gait versatility' (L35) seem to indicate that no firm conclusion can be drawn, and that they fit somewhere in between the extremes, and so the conclusion of convergent evolution is not warranted.

L32: 'distinct amongst' - 'distinct from' is clearer.

L77: change 'inferring' to 'implying'.

L136: does 'photographed from the right-hand side' refer to lateral or medial direction, as this depends on whether the bone is from the right or left side of the animal?

L157: the wording 'data became insufficient' is unclear given that you are increasing the number of PCs included in the analysis.

L183: references to figures would help here in describing the anatomical details.

L254: ML is nearly parallel to WTS in the figure, but is listed separately here. LLTC and LMTC fall intermediate to the other groups.

L257: does 'dimensionally compatible' mean approximately the same size?

L271: the latter part of this paragraph is more Discussion than Results.

L301: correct spelling of 'Hypsiprymnodon'. Also L303, L399.

L304: 'far more advanced scansorial ... QM' - the end of the Results, and data in Table 1, show that these fossils were as much 'terrestrial/bipedal saltating' than 'arboreal/scansorial', which calls into question this line of conclusions of arboreality, either primitively or independently evolved. Likewise L331 'advanced scansorial morphologies'.

L518: change to 'dependent'.

Reviewer: 2

Comments to the Author(s)

Overall I believe this is a rigorous analysis and the manuscript is clear and well written. I have only a few relatively minor comments that I feel should be clarified.

1 - Many of the conclusions drawn from this study are supported by data in the supplementary files. Why were these figures/tables not included in the main manuscript?

2 - In figure 1, LTC is not defined.

3 - In figures 2-4, I was confused by the outlines of macropods adjacent to the figure legends. What is the significance of these?

4 - I expect that it will not be an issue in publication, but in my printed copy, the symbols in figures 2-4 were difficult to distinguish.

Author's Response to Decision Letter for (RSOS-181617.R0)

See Appendix A.

Decision letter (RSOS-181617.R1)

09-Jan-2019

Dear Dr Kear,

I am pleased to inform you that your manuscript entitled "Climbing adaptations, locomotory disparity, and ecological convergence in ancient stem 'kangaroos'" is now accepted for publication in Royal Society Open Science.

Royal Society Open Science operates under a continuous publication model (<http://bit.ly/cpFAQ>). Your article will be published straight into the next open issue and this will be the final version of the paper. As such, it can be cited immediately by other researchers.

As the issue version of your paper will be the only version to be published I would advise you to check your proofs thoroughly as changes cannot be made once the paper is published.

on behalf of Dr Monica Daley (Associate Editor) and Kevin Padian (Subject Editor)
openscience@royalsociety.org

Follow Royal Society Publishing on Twitter: [@RSocPublishing](https://twitter.com/RSocPublishing)

Appendix A

Responses to Reviewers

Associate Editor Comments to Author (Dr Monica Daley):

Associate Editor: 1

Comments to the Author:

Two expert reviewers have evaluated your paper interpreting locomotor modes of macropodoids from pedal morphometrics. The reviewers suggest that the work makes a scientifically sound contribution, but they also highlight the ambiguity of the results, and suggest that some of the conclusions drawn in the text should be revised to be better aligned with this ambiguity. I will consider a revised version of this paper that fully addresses the reviewers comments to make sure that all of the main findings and conclusions are fully supported by the data presented within the main text.

We would like to thank both the editors and reviewers for their assistance in improving our manuscript. We have responded to each of their comments in turn below.

Reviewer comments to Author:

Reviewer: 1

Comments to the Author(s)

This manuscript aims to interpret the locomotor mode of early Macropodiformes from pedal morphology through morphometric comparisons with modern species. The authors examine some of the best preserved materials for stem macropodoids and use both traditional linear metrics as well as 2D geometric morphometrics of bones.

We again thank Reviewer 1 for highlighting both the quality of our fossil material, and the multiple cross-referencing analyses that we have used to assess our data.

The analyses are clearly set out, and often do not give a clear answer to the questions asked.

We set out to determine the possible range of balbarid (and other basally branching fossil macropodoid) locomotion using the best available fossils, and to “test the potential for hopping” (L94-95). Our analyses produced contrasting results, but as we explain, this is entirely consistent with the morphological variability observed in our fossils, and across the spectrum of extant macropodoids.

This may be because these earlier forms are less derived or more generalised than the majority of modern macropodoids.

We would point out that our fossils represent two quite distinct tarsal and pedal morphotypes that are by no means ‘generalised’ or ‘less derived’ than those of extant macropodoids. Rather as our results show, these fossils manifest character states, morphologies and proportions that are

highly derived amongst macropodoids generally (discussed in detail on L191-208), and are as different from each other in many respects (described on L199-206), as they are from living rat-kangaroos, wallabies and kangaroos. Indeed, this is one of the key findings of our study, and is repeatedly emphasised in the Abstract (L32-34), Results (L199-208, L226-229, L231-232, L256-261, L262-271), and at length in the Discussion (L308-325). We also explicitly explain that these features are independently elaborated relative to more basally branching macropodoids, such as *Hypsiprymnodon moschatus* (which is terrestrial but able to climb over uneven terrain: Burk et al. 1998. *Syst. Biol.* **47**, 457–474.), and thus demonstrate an unambiguous example of character state convergence compared to modern macropodine tree-kangaroos (L297-306).

The key result is the BLR summary (Figure 5) that gives the mean position of both fossils at ~0.70-0.95 along the locomotor gradient towards the terrestrial/bipedal saltating end. However, the main conclusions in the text do not follow this finding, and suggest that arboreal/scansorial is more likely (L304, L331) and represents a convergent adaptation to arboreality (L37; Figure 6).

Nowhere do we conclude that our BLR results “suggest that arboreal/scansorial is more likely”. On the contrary, our BLR predictions as summarised in the previous figure 5 depict the “range of predicted values for the fossils” (previous L530-531), which although skewed towards the “terrestrial/bipedal saltating” bin, reveal a spectrum of results from the astragalar (~0.1-0.99), and pedal digit IV ungual datasets (~0.4-0.98). This is clearly evident in the BLR results from each separate dataset shown in the previous ESM figures S2, S3 (now figures 5 and 6). Our “main conclusions” are explicitly stated as being “morphological categorisation of QM F59022, QM F59025, and *N. gillespieae* with terrestrial/bipedal saltating macropodoids” (L275-276). We also note that some of our datasets “indicated arboreal/scansorial habits, especially for QM F59022 and QM F59025” (L279-280), and that this “demonstrates ambiguity in our ability to precisely estimate ecological and locomotory capabilities, but also highlights the considerable novel anatomical variation exhibited by these ancient stem macropodoids, even within single bones” (L281-283).

Furthermore, based on comparisons with extant macropodoids, we conclude that our results are suggestive of gait variability, and NOT an exclusive indicator of “arboreal/scansorial” habits (L308-325). In fact, we have been very careful to avoid any overreaching claims of “arboreality”, and repeatedly emphasise this by our use of the alternative term “scansoriality” (see previous L37, L304, L331, and elsewhere). Scansoriality describes the ability to climb, NOT arboreality per se, which in our fossils is evidenced by tarsal and pedal character states that resemble those of modern tree kangaroos, but are undoubtedly independently derived.

The results and conclusions should more closely align to allow sufficient support. Statements such as ‘archetypal gait versatility’ (L35) seem to indicate that no firm conclusion can be drawn, and that they fit somewhere in between the extremes, and so the conclusion of convergent evolution is not warranted.

Reviewer 1 has misinterpreted our results and conclusions. The phrase “archetypal gait versatility” (L35) explains that morphological characteristics consistent with versatile gaits are observable even in the most ancient fossils. “Convergent evolution” IS evident in the numerous character states shared by our fossils and modern macropodine tree-kangaroos, which constitute a separate evolutionary lineage stratigraphically segregated by up to 10 Ma. We have

deliberately avoided any suggestion that ancient macropodoids habitually lived in trees (vis-a-vis “arboreality”), as implied by Reviewer 1. Rather, we make the point that their independent acquisition of morphologically similar tarsal and pedal traits is consistent with climbing (= scansoriality), and probably evinces one amongst several non-mutually exclusive gait modes (L35-37, L239-251, L308-325).

We have modified the following lines to clarify our interpretations: L35-37 “probably integrated higher-speed hopping with slower-speed quadrupedal progression, and varying degrees of scansoriality as independent specialisations for life in woodland and forest settings”; L103-104 “novel evidence of independently specialised morphologies”; L338-339 “Specialisation towards scansorial lifestyles could therefore have proven similarly detrimental”.

L32: ‘distinct amongst’ – ‘distinct from’ is clearer.

Changed.

L77: change ‘inferring’ to ‘implying’.

Changed.

L136: does ‘photographed from the right-hand side’ refer to lateral or medial direction, as this depends on whether the bone is from the right or left side of the animal?

Rewritten to: “all left pedal digit IV unguals were photographed from the right-hand (medial) side”.

L157: the wording ‘data became insufficient’ is unclear given that you are increasing the number of PCs included in the analysis.

Rewritten to: “data failed to produce a reliable BLR result”

L183: references to figures would help here in describing the anatomical details.

Added.

L254: ML is nearly parallel to WTS in the figure, but is listed separately here. LLTC and LMTC fall intermediate to the other groups.

Rewritten to: “WTS and ML versus HD and HNF, and WNF versus MW, LMTC and LLTC”.

L257: does ‘dimensionally compatible’ mean approximately the same size?

No. We are describing correlated variable loadings in PC2 and PC3, and thus exclude the size-related vector PC1. Rewritten to: “compatible with *Dendrolagus*”.

L271: the latter part of this paragraph is more Discussion than Results.

See previous comment. This paragraph is retained to avoid miss-interpretation of our results.

L301: correct spelling of ‘Hypsiprymnodon’. Also L303, L399.

Corrected.

L304: ‘far more advanced scansorial ... QM’ – the end of the Results, and data in Table 1, show that these fossils were as much ‘terrestrial/bipedal saltating’ than ‘arboreal/scansorial’, which calls into question this line of conclusions of arboreality, either primitively or independently evolved. Likewise L331 ‘advanced scansorial morphologies’.

Again, we would point out that “arboreal” (=living in trees) and “scansorial” (=adapted for climbing) have different meanings. We use “scansorial” to describe the ability to climb without any implication of predominantly arboreal lifestyles.

L518: change to ‘dependent’.

Corrected.

Reviewer: 2

Comments to the Author(s)

Overall I believe this is a rigorous analysis and the manuscript is clear and well written. I have only a few relatively minor comments that I feel should be clarified.

We thank Reviewer 2 for their positive summary of our analysis and manuscript.

1 - Many of the conclusions drawn from this study are supported by data in the supplementary files. Why were these figures/tables not included in the main manuscript?

We have replaced the previous figure 5 with the supplemental figures S2 and S3 (now figures 5 and 6), which depict the same results but more clearly demonstrate the variation in our BLRs.

2 - In figure 1, LTC is not defined.

Corrected.

3 - In figures 2-4, I was confused by the outlines of macropods adjacent to the figure legends. What is the significance of these?

These were included to provide graphical representations of the different macropodoid body-forms, and thus visually depict a generalised balbarid (figure 1), tree-kangaroo (figure 2), and terrestrial hopping kangaroo (figure 3). This has now been mentioned in the figure captions to clarify.

4 - I expect that it will not be an issue in publication, but in my printed copy, the symbols in figures 2-4 were difficult to distinguish.

These have been enlarged in the final versions of the figures.